# Pretreatment of Melanoma Cells with Aqueous Ethanol Extract from *Madhuca longifolia* Bark Strongly Potentiates the Activity of a Low Dose of Dacarbazine

**DOI:** 10.3390/ijms25137220

**Published:** 2024-06-29

**Authors:** Kamila Środa-Pomianek, Anna Barycka, Michał Gleńsk, Meena Rajbhandari, Magdalena Skonieczna, Anna Palko-Łabuz, Olga Wesołowska

**Affiliations:** 1Department of Biophysics and Neurobiology, Wroclaw Medical University, 50-369 Wrocław, Poland; kamila.sroda-pomianek@umw.edu.pl (K.Ś.-P.); abarycka79@gmail.com (A.B.); 2Department of Pharmacognosy and Herbal Medicines, Wrocław Medical University, 50-367 Wrocław, Poland; michal.glensk@umw.edu.pl; 3Research Centre for Applied Science and Technology (RECAST), Kirtipur 44600, Kathmandu, Nepal; karmacharyameena@gmail.com; 4Department of Systems Biology and Engineering, The Silesian University of Technology, 44-100 Gliwice, Poland; magdalena.skonieczna@polsl.pl; 5Biotechnology Centre, Silesian University of Technology, 44-100 Gliwice, Poland

**Keywords:** *Madhuca longifolia*, bark, ethanol extract, dacarbazine, melanoma, cytotoxicity, potentiation of anticancer activity

## Abstract

*Madhuca longifolia* is an evergreen tree distributed in India, Nepal, and Sri Lanka. This tree is commonly known as Mahua and is used in traditional medicine. It was demonstrated that ethanol extract from the bark of *M. longifolia* possessed potent cytotoxic activity towards two melanoma cell lines, in contrast to aqueous extract that exhibited no activity. Apart from being selectively cytotoxic to cancer cells (with no activity towards non-cancerous fibroblasts), the studied extract induced apoptosis and increased reactive oxygen species generation in melanoma cells. Additionally, the use of the extract together with dacarbazine (both in non-toxic concentrations) resulted in the enhancement of their anticancer activity. Moreover, the pretreatment of melanoma cells with *M. longifolia* extract potentiated the activity of a low dose of dacarbazine to an even higher extent. It was concluded that ethanol extract of *M. longifolia* sensitized human melanoma cells to chemotherapeutic drugs. It can therefore be interesting as a promising source of compounds for prospective combination therapy.

## 1. Introduction

*Madhuca longifolia* var. *longifolia* is an evergreen tree belonging to the family Sapotaceae, distributed in India, Nepal, and Sri Lanka. This tree is commonly known as Mahua and is used traditionally in the treatment of many diseases including rheumatism, epilepsy, ulcers, and inflammations [1,2]. Traditionally the bark of *M. longifolia* is considered to be beneficial in treating itch, swelling, and bites of poisonous snakes. It has been used internally in diabetes, stomach ulcers, tonsillitis, and pharyngitis [1]. Among active compounds that were previously detected and confirmed in *Madhuca* the following might be listed: triterpenoids, saponins, steroids, flavonoids and their glycosides, and tannins [3,4,5]. It has been demonstrated that the extracts from *M. longifolia* possessed antioxidant, anti-inflammatory, antibacterial, antiepileptic, antipyretic, and anticancer properties [6,7,8,9,10,11].

Melanoma development originates from malignant transformation of melanocytes. It accounts for c.a. 1–2 % of malignant tumors but its incidence is growing [12]. While the treatment of melanoma in situ is usually successful, the 5-year survival rate for patients with metastatic melanoma is only 20–30%. That is why melanoma ranks in the top ten cancers characterized by the highest mortality. The introduction of immunotherapies in recent years led to significant advancement in treatment for advanced melanoma [12]. Traditional melanoma chemotherapy with alkylating agents such as dacarbazine and temozolomide has been largely replaced by new therapies. However, there are some patients with refractory or relapsed disease for whom chemotherapy is still used as a palliative option. Its use of alkylating agents is accompanied by serious side effects such as nausea and vomiting, flu-like symptoms, hepatotoxicity, neurotoxicity, and suppression of hematopoiesis [13]. This limits drug dose and reduces its efficacy.

In the present work, the anticancer properties of aqueous and ethanol extract from the bark of *Madhuca longifolia* were investigated in human melanoma cell lines. It was demonstrated that the ethanol extract potently limited cancer cell viability, promoted apoptosis, and increased the level of reactive oxygen species (ROS) in melanoma cells. In addition, the ethanol extract from *M. longifolia* was tested in combination with dacarbazine. It was proved that the simultaneous application of the extract with the drug maintained the high anticancer effectiveness of dacarbazine while lowering its dose to 1 ug/mL, which was c.a. 10 times lower than IC_50_ values reported for dacarbazine in various melanoma cell lines [14,15]. Moreover, the efficacy of *M. longifolia* extract in potentiating anticancer drug activity was strongly increased when cancer cells were preincubated with the extract prior to dacarbazine treatment. It was therefore concluded that the studied extract possessed the property of sensitization of human melanoma cells to chemotherapeutic drugs and thus might be interesting as a source of compounds for prospective combination therapy.

## 2. Results

### 2.1. Analysis of Madhuca longifolia Extracts

Nine distinctive peaks were observed in LC-MS spectra of aqueous ethanol extract from the bark of *M. longifolia* (MLE). The data are presented in Figure 1 and the Appendix A. These peaks were tentatively characterized based on their MS and MS/MS spectra.

When the spectrum of peak (1) was analyzed, the pseudomolecular ion appeared [M − H]^−^ at m/z 137.0245, suggesting the formula C_7_H_5_O_3_. This formula was in good agreement with hydroxybenzoic acid derivatives such as *p*-hydroxybenzoic acid or salicylic acid.

Peak (2) was characterized by the pseudomolecular ion [M − H]^−^ 577.1357 with formula C_30_H_25_O_12_. Moreover, in the MS/MS experiment, fragment ions at m/z 407, m/z 289 and m/z 125 were observed. Additionally, an ion at m/z 865.1994 of smaller intensity was observed. Therefore, peak (2) was tentatively identified as a mixture of procyanidins (dimer and trimer).

Peak (3) of the highest intensity in LC-MS chromatogram was characterized by another pseudomolecular ion [M − H]^−^ 577.1357 with formula C_30_H_25_O_12_. Moreover, in the MS/MS experiment, ions at m/z 407, m/z 289, and m/z 125 were observed. Therefore, peak (3) was tentatively identified as a procyanidin dimer.

Peak (4) in the MS spectra gave a predominant ion [M − H]^−^ 289.0723 (C_15_H_13_O_6_), together with the fragment ions at m/z 245, m/z 151, and m/z 123, suggesting the presence of catechin/epicatechin molecule. Moreover, two minor signals were observed [M − H]^−^ at m/z 577.1356 and 865.2004, which are consistent with dimer and trimer procyanidins, respectively.

The following peak (5) gave a major ion [M − H]^−^ 409.0816 (was not assigned) and two ions of low intensity at m/z 865.2025 and 1153.2631, which belong to procyanidins.

Peak (6) was characterized by the predominant ion [M − H]^−^ 561.1402 (C_30_H_25_O_11_) with fragments at m/z 407 and m/z 289, suggesting the presence of a procyanidin dimer. Additionally, ion [M − H]^−^ at m/z 865.1994 with the fragment ions at m/z 713, m/z 577, m/z 425, and m/z 287, typical for procyanidin trimers, was present.

Peak (7) in the MS spectrum showed the main pseudomolecular ion [M − H]^−^ at m/z 425.0763 and was not assigned to procyanidin due to its different fragmentation pattern. Its MS/MS fragmentation showed two intense peaks at m/z 241 and 96 and was tentatively characterized as 3,4,5-trimetoxyphenyl 1-*O*-β-(6-sulpho)-glucopyranoside [16].

Peak (8) of the low intensity in the LC-MS chromatogram showed three pseudomolecular ions [M − H]^−^ at m/z 577.1356, 865.2006, and 1153.2589, masses that are typical for procyanidin dimers, trimers, and tetramers.

At the MS spectrum of peak (9), the main ion [M − H]^−^ at m/z 503.3384 (C_30_H_47_O_6_) was recorded together with fragment ions at m/z 485 and m/z 459. As previously [17], saponins with protobassic acid aglycone were isolated from the seeds of *Madhuca longifolia*, and we tentatively characterized this peak as a protobassic acid.

The masses of all procyanidins were within 3 ppm (∆ ppm) of their molecular formulas and were in good agreement with those previously reported by Lin et al. [18] and Bürkel et al. [5].

In the case of aqueous extract (MLA), the general LC-MS profile of phenolic compounds is similar to ethanol extract; however, there are differences in the intensity of ndividual compounds. Additionally, in the aqueous extract, there was no peak that belonged to triterpenoids.

### 2.2. Influence of the Extracts on Cell Viability

The analysis of the influence of MLE and MLA on the viability of human melanoma cells revealed that MLE was potently cytotoxic to 1205-Lu and Me45 cells (Figure 2A and B, respectively) during 48 h treatment. In contrast, MLA did not change the viability of 1205-Lu cells and even stimulated the growth of Me45 cells, which effectively vanished only at the concentration of 200 μg/mL. It could be noticed that 1205-Lu cells were more vulnerable to the presence of MLE than Me45 cells. It was also confirmed by the estimation of IC_50_ values of MLE. This parameter was found to be 2.57 ± 0.36 μg/mL in 1205-Lu cells and 15.13 ± 1.02 μg/mL in Me45 cells. In the case of MLA, it was impossible to find its IC_50_ in the studied range of concentrations. In contrast to melanoma cells, the viability of human neonatal dermal fibroblasts (Figure 2C) was only slightly affected by either MLE or MLA. The inhibition of cell growth never exceeded 20% when the studied extracts were applied in concentrations up to 200 μg/mL. The results of the MTT assay were also corroborated by microscopic analysis of melanoma and NHDF cells treated with 100 μg/mL of MLE or MLA (Figure 3). It was concluded that MLE was significantly more cytotoxic to cancerous cells as compared to non-cancerous cells.

### 2.3. Influence of MLE on Cell Cycle

Dacarbazine is an alkylating agent used in the treatment of metastatic melanoma. Its use is accompanied by serious side effects such as the inhibition of the hematopoietic activity of the bone marrow, which limits the drug’s dose. In the present work, we aimed to test the possibility of using *M. longifolia* extracts together with this drug, thus supporting its anticancer activity and reducing the required dose. For further experiments, only MLE was chosen since MLA did not exhibit any cytotoxicity towards human melanoma cells. Both dacarbazine and MLE were used in the concentrations that caused no effect by themselves, i.e. dacarbazine at 1 μg/mL and MLE at 0.1 μg/mL. The influence of MLE on the cell cycle of cancer cells was studied using flow cytometry. The cells were assigned to the appropriate cycle phases on the base of their DNA content. As expected, neither the drug nor MLE affected 1205-Lu and Me45 cells significantly (Figure 4). When both MLE and the drug were applied together, the increased number of dead (fragmented) cells was observed in both melanoma cell lines, with a concomitant decrease in other cell populations. Moreover, if the cells were preincubated with MLE (0.1 μg/mL) for 4 h before adding dacarbazine, the effect of the mixture of melanoma cells was clearly stronger than the effect exerted with no preincubation. Additionally, the incubation of melanoma cells with MLE at 0.1 μg/mL for 52 h (4 h + 48 h) did not change the distribution of melanoma cells between different cell cycle phases much. The only exception was observed in Me45 cells in which the slightly increased number of the cells in the G_2_/M phase was recorded. It was concluded that the pretreatment of melanoma cells with MLE in low concentrations in some way sensitized cancer cells to dacarbazine.

### 2.4. Induction of Apoptosis

Next, apoptosis induced by the studied chemicals was investigated using a flow cytometric assay in which the presence of phosphatidylserine in the outer membrane monolayer is detected via Anexin-V binding. Exemplary histograms are presented in Figure 5. The results obtained in 1205-Lu and Me45 cells were similar (Figure 6). Dacarbazine at 1 ug/mL slightly increased the number of apoptotic cells. Similar activity was also noticed for MLE when used at a concentration equal to its IC_10_ value. When the concentration of the extract was increased to IC_50_ value, its proapoptotic activity also became stronger. Similarly, as in the previous experiment, the simultaneous application of dacarbazine and MLE in low concentrations (1 ug/mL and IC_10_, respectively) resulted in the reduction of the normal cell population associated with the increase in apoptotic and necrotic cell numbers. When cancer cells were preincubated with the extract for 4 h prior to the experiment, higher proapoptotic activity was noticed as compared to simultaneous treatment by both MLE and dacarbazine.

Apoptosis-induction potency of ethanol extract of *M. longifolia* was also investigated by studying the activation of the proteolytic enzyme, caspase-3, which is associated with the execution phase of apoptosis. Both in 1205-Lu and in Me45 cells, MLE caused concentration-dependent growth in caspase-3 activity (Figure 7). Dacarbazine at the concentration of 1 μg/mL also slightly increased the activity of this enzyme in melanoma cells. When cancer cells were treated first with MLE at 0.1 μg/mL (for 4 h) and then dacarbazine was added and incubation was continued for 48 h, the activity of caspase-3 was c.a. 2.5-fold higher as compared to untreated cells.

Additionally, dose and effect data obtained from caspase-3 assay for pure compounds and for two-component combinations (decarbazine/MLE) were subjected to CompuSyn analysis (Table 1). The proapoptotic effects of dacarbazine and MLE were synergistic as demonstrated by the values of the combination index (CI) that were well below 1 for MLE used at concentrations of 0.05, 0.1, and 0.5 μg/mL). On the other hand, no synergy was detected between MLE and the drug when the extract was used at a concentration of 5 μg/mL when the effect was purely additive.

### 2.5. Influence of MLE on Intracellular ROS Level

Next, the generation of ROS in human melanoma cells treated by MLE, dacarbazine, or both was quantified. The results are presented in Figure 8. We noticed that all the treatments resulted in an increase in ROS levels both in 1205-Lu and Me45 cells. The 1205-Lu cells seemed, however, more sensitive to the studied chemicals since the range of the observed changes was larger there as compared to Me45 cells. Dacarbazine (at 1 μg/mL) slightly increased ROS level, while the activity of MLE (at 0.1 μg/mL) in this respect was rather weak. On the other hand, a strong increase in intracellular ROS content was observed if both the drug and the extract were applied together, and the increase was further potentiated when cancer cells were pretreated with MLE before the application of the anticancer drug. It was concluded that the presence of MLE enhanced the prooxidative action of dacarbazine, especially if melanoma cells were pretreated with ethanol extract from *M. longifolia* bark.

## 3. Discussion

In the present work, the high anticancer activity of the ethanol extract from the bark of *Madhuca longifolia* was demonstrated. In contrast, aqueous extract possessed no activity against cancer cells. The presence of plenty of potentially biologically active compounds has been confirmed in various parts of *M. longifolia*. The active compounds belonged to such groups as triterpenoids, saponins, steroids, flavonoids and their glycosides, and tannins [3,4,5]. Although tannins have been previously reported in the bark of *Madhuca longifolia*, the present work, together with the report of Burkel et al. [5], is the first to demonstrate the presence of procyanidins in alcoholic extracts from the bark of this plant. Taking into account the well-characterized anticancer properties of procyanidins [19,20,21,22], it was hypothesized that procyanidins were likely to be the main compounds responsible for the observed activity. However, the presence of a pentacyclic triterpenoid in the ethanol extract could also have an impact on the overall activity, especially in terms of cytotoxicity. Additionally, the use of a plant extract instead of its purified active constituents might have such an advantage that the presence of multiple active compounds could lead to mutual potentiating and modulating of their activity that would not happen when any single compound is used [23].

The cytotoxicity of MLE to human melanoma cells was significant, and the extract was more toxic to 1205-Lu than to Me45 cells. In contrast, the studied extract was only slightly cytotoxic to normal human fibroblasts in concentrations up to 200 μg/mL. This suggested some degree of selectivity of MLE towards malignant cells. Ethanol extract from the bark of *M. longifolia* has been previously demonstrated to be potently cytotoxic to breast cancer cells [9], and its impact on normal cells was negligible. Significant anticancer properties were also observed for extracts from other parts of the *M. longifolia* plant, namely leaves [11] and seeds [24]. The closely related plant, *M. indica*, was also the source of anticancer compounds, as ethanol extract of the whole plant turned out to be cytotoxic to neuroblastoma, lung, and colon cancer cells [7]. Additionally, Yadav et al. [25] showed that the application of gold nanoparticles loaded with *M. longifolia* bark extract resulted in anti-melanoma efficacy in tumor-bearing mice.

Further experiments demonstrated that MLE promoted apoptosis in human melanoma cells. When the extract was applied at a concentration equal to its IC_50_ value, the number of apoptotic cells was approaching 50% in both melanoma cell lines. Additionally, MLE increased the activity of caspase-3 in a concentration-dependent manner. To obtain a full picture of the situation it should be added that MLE at very low concentration (0.1 μg/mL) slightly increased the intracellular level of ROS in cancer cells. The above observations shed some light on the putative mechanism of anticancer activity of MLE. A similar mechanism of cancer cells’ death induction was also reported previously for methanolic extract of *M. longifolia* leaves [10].

The goal of the present work was testing MLE as a putative supplement to dacarbazine that could increase its anticancer activity and reduce toxicity at the same time. The dose of dacarbazine used in the experiments was c.a. 10 times lower than its IC_50_ values recorded in melanoma cells [14,15]. The drug used at 1 μg/mL did not affect the cell cycle of melanoma cells, but, similarly to MLE, slightly increased ROS level and induced apoptosis. When dacarbazine was combined with MLE, an increase in the number of apoptotic and dead melanoma cells, as well as in intracellular ROS production, was observed. Additionally, the synergistic interaction between MLE and dacarbazine was demonstrated by combination index analysis based on caspase 3 assay. Since both the drug and MLE were likely to affect similar cellular mechanisms, the effect of the pretreatment of melanoma cells with MLE before the application of dacarbazine was additionally tested. When MLE was applied 4 h before the drug, and then the incubation was continued further for 48 h, the potentiation of the anticancer effect in human melanoma cells was observed. In both 1205-Lu and Me45 cells, the number of sub-G_1_ phase cells (dead, fragmented) significantly increased, while populations of the cells in other cell cycle phases were reduced. Additionally, the size of the apoptotic cell population was significantly larger than in the case of dacarbazine and MLE applied simultaneously. The two types of treatment differed also in their influence on intracellular ROS production. The pretreatment of melanoma cells with MLE resulted in more than 2-fold higher ROS level as compared to the situation when both additions were given together. Therefore, it seemed clear that preincubation of cancer cells with MLE in some way sensitized them to the action of the anticancer drug. The observed effect is somehow similar to the one observed by Riganti et al. [26]. The authors reported that two low doses of doxorubicin repeated in 24 h interval were more cytotoxic to colon cancer cells than a single dose of 5 times higher concentration. It was explained by the higher ROS production induced by the first type of treatment. Taking into account the great difference between ROS levels observed in the present work for the simultaneous dacarbazine and MLE application and MLE pretreatment observed in the present work, it was assumed that a similar mechanism might be responsible for the potentiation of dacarbazine anticancer activity by *M. longifolia* bark extract.

The strong antioxidant activity of *M. longifolia* extracts has been previously reported [9,11]. This may seem to be in contradiction with the results obtained in the present work, where prooxidant activity of MLE was noticed. The apparent discrepancy might result from the different methodologies used in the studies. Antioxidant properties were reported in the studies in which simple model systems were used like DPPH assay [9], various types of radicals scavenging tests, or lipid peroxidation assay [11]. In contrast, in the present work whole cells were used for the experiments, and ROS generation was monitored. Similarly, Sarkar et al. [10] with the extract from *M. longifolia* leaves.

## 4. Materials and Methods

### 4.1. Chemicals

Dacarbazine and camptothecin were purchased in Sigma-Aldrich (Poznan, Poland), dissolved in dimethyl sulfoxide (DMSO), and stored at −20 °C. Ethanol (MLE) and aqueous (MLA) extracts from *Madhuca longifolia* bark were dissolved in DMSO/water (1:1) mixture. Stock solutions of the studied compounds were diluted in a cell culture medium just before the experiments.

### 4.2. Plant Material

The plant material (bark of *Madhuca longifolia*) was collected in August 2016 in the Kailali district of western Nepal and the voucher specimens (ML_01) have been deposited in the Herbarium of Department of Pharmacognosy and Herbal Medicines, Wroclaw Medical University, Wroclaw, Poland.

### 4.3. Extraction

The bark of *M. longifolia* was dried, ground into a fine powder with a grinder (Ika A11, Staufen, Germany), and extracted with 70% ethanol at room temperature (MLE) or with hot distilled water (MLA) for 24 h. The drug-to-solvent ratio was 1 to 10 (g/mL). After the removal of remaining solvents on a rotavapor R-210 (Buchi, Flavil, Switzerland), the residues were dried in a vacuum chamber for the next 24 h.

### 4.4. UHPLC-ESI-MS and MS/MS Analysis

For UHPLC-ESI-MS analysis samples were dissolved in methanol or water, respectively (1 mg of the sample per 1 mL), filtered through the 0.22 μm PTFE syringe filter (Merck-Millipore, Darmstadt, Germany), and stored at 4 °C before the analysis.

Analytical UHPLC separation was conducted on the Thermo Scientific UHPLC Ultimate 3000 apparatus (Thermo Fisher Scientific, Waltham, MA, USA) consisting of an LPG-3400RS quaternary pump with a vacuum degasser, a WPS-3000RS autosampler, and a TCC-3000SD column oven. The system was additionally integrated with an ESI-qTOF Compact HRMS detector (Bruker Daltonics, Bremen, Germany). Separations were performed on a Kinetex RP-18 column (100 × 2.1 mm × 2.6 μm; Phenomenex, Torrance, CA, USA). The UHPLC-ESI-MS system was operated in the negative mode and calibrated with the TunemixTM mixture (Agilent Technologies, Palo Alto, CA, USA). Data Analysis software version 5.3 (Bruker Daltonics, Bremen, Germany) was utilized for data collection and evaluation of the obtained mass spectra. The main instrumental parameters were scan range: 50–2200 m/z; dry gas: nitrogen; temperature: 200 °C; potential between the spray needle and the orifice: 4.2 kV. Collision energy in CID cells was 35 eV. The applied chromatographic gradient solvent system consisted of solvents A (0.1 % HCOOH in water) and B (0.1 % HCOOH in acetonitrile). The injection volume was 2 μL and the flow rate was 0.3 mL/min. The following elution program was used: 0→30 min (5→95% B), 30→40 min (95% B), 40→45 min (95→5% B), and 45→50 min (5% B). UHPLC analyses were carried out isothermally at 30 °C.

### 4.5. Cell Culture

Malignant melanoma cell lines (Me45 and 1205-Lu) were provided by the collection of the Centre of Oncology (Gliwice, Poland), and human neonatal dermal fibroblasts (NHDF) were purchased in Lonza (Basel, Switzerland). The Me45 cell line was established in 1997 at the Radiobiology Department of the Centre of Oncology and it was derived from a metastasis of skin melanoma to a lymph node. The 1205-Lu cell line originates from human melanoma metastases to the lungs of immunodeficient mice [27]. All types of cells were grown in DMEM-F12 medium (CytoGen, Pabianice, Poland) supplemented with 10% fetal bovine serum (Gibco, Waltham, MA, USA), 1% L-glutamine (Sigma-Aldrich, Poznan, Poland), and 1% of antibiotics (10,000 μg/mL streptomycin and 10,000 units/mL penicillin; Sigma-Aldrich, Poznan, Poland). Cells were cultivated in a humidified atmosphere (37 °C, 5% CO_2_).

### 4.6. Cytotoxicity Assay

The cells were seeded on 96-well plates (20,000 cells/mL) and left overnight to firmly attach. Then, the fresh portion of the medium containing appropriate concentrations of the studied extracts was added and the incubation was continued for 48 h (37 °C, 5% CO_2_). The controls containing only the solvent were also prepared. After the removal of the medium, the plates were washed with physiological saline (PBS; PAA, Warsaw, Poland), and incubated with MTT (3-[4,5-dimethylthiazol-2-yl]-2,5-diphenyltetrazolium bromide, Sigma-Aldrich, Poznan, Poland) at the concentration of 0.5 mg/mL for 2 h. Next, the MTT solution was discarded, and the formazan crystals were let to dissolve in the mixture of isopropanol/HCl (*v*/*v* 1:0.04; Sigma-Aldrich, Poznan, Poland). The absorbance was read at 570 nm using a microplate reader (Tecan, Maennedorf, Switzerland). Cell viability was calculated from the ratio (A_570_ of treated cells/A_570_ of control cells) × 100%. Concentrations required to reduce cell number to 50% (IC_50_ values) were determined with the use of CompuSyn software (version 1.0, ComboSyn Inc., Paramus, NJ, USA). All experiments were performed in triplicate. The photographs of cells were collected with the use of a Nikon Eclipse TE2000-E inverted microscope (Nikon Instruments, Amstelveen, The Netherlands) with a PlanFluor objective (10×, NA = 0.3).

### 4.7. Flow Cytometry

Flow cytometry method was employed to investigate cell cycle, apoptosis, and intracellular level of ROS. The cells were seeded onto 12-well plates and prepared in the same way as for the MTT assay. After incubation, the cells were harvested and centrifuged (1500 rpm, 3 min). For cell cycle analysis, the cells were fixed with hypotonic buffer (PBS with 5 mg/L of citric acid; 1:9 Triton-X solution; RNase 100 µg/mL) containing propidium iodide (PI; Sigma-Aldrich, Poznan, Poland) at 100 µg/mL. Then, the samples were incubated for 20 min at room temperature in darkness and stored on ice until the measurement.

For apoptosis detection, Annexin-V apoptosis assay (BioLegend, San Diego, CA, USA) was used. Briefly, the harvested cells were washed with PBS and centrifuged (1500 rpm, 3 min). The pellet was suspended in 50 μl of cold Annexin-V binding buffer and stained with FITC-labeled Annexin-V for 30 min at 37 °C in darkness. Next, Annexin-V binding buffer and PI solution (100 μg/mL) were added and the probes were kept on ice until they were measured.

During the experiments on ROS content, the harvested cells were washed with PBS and centrifuged (1500 rpm, 3 min). The pellet was suspended in DMEM-F12 medium and a cell-permeable non-fluorescent probe 2′,7′-dichlorofluorescin diacetate (DCFH-DA, Sigma-Aldrich, Poznan, Poland) was added at a final concentration of 30µM. After incubation (30 min, 37 °C), the probes were stored on ice until the measurement. During the incubation, DCFH-DA is hydrolyzed by cellular esterases into DCFH, which in turn undergoes oxidation and turns into highly fluorescent 2′,7′-dichlorofluorescein (DCF) due to the presence of intracellular ROS and other peroxides.

Aria III flow cytometer (Becton Dickinson, Franklin Lakes, NJ, USA) with FITC configuration (488 nm excitation; emission: LP mirror 503, BP filter 530/30) or with PE configuration (547 nm excitation; emission: 585 nm) was employed for the analysis. All experiments were repeated 3 times, and at least 10,000 cells were counted each time.

### 4.8. Caspase 3 Acitvity

A commercially available kit (GenScript Biotech, Leiden, the Netherlands) was used to monitor the activity of caspase-3. The cells were seeded onto 6-well plates and prepared in the same way as for the MTT assay. After 48 h incubation with studied extracts (37 °C, 5% CO_2_), the cells were scraped, centrifuged (2000× *g*, 5 min, 25 °C), and lysed. Spectrophotometric detection (A_405_) of the chromophore *p*-nitroanilide (pNA) was used to evaluate caspase-3 activity. The relative increase in caspase-3 activity was determined by comparing the absorbance of pNA in the studied sample and the control (untreated sample). Experiments were performed in triplicate.

### 4.9. Combination Index Analysis

The combination index (CI) was calculated using CompuSyn software (version 1.0, ComboSyn Inc., Paramus, NJ, USA) according to the classic median-effect Equation (1), as was previously described by [28].
(1)$$CI=\frac{{\left(D\right)}_{1}}{{\left(Dx\right)}_{1}}+\frac{{\left(D\right)}_{2}}{{\left(Dx\right)}_{2}}$$

In Equation (1), (Dx)_1_ is the dose of drug 1 alone that affects a system by x%, (Dx)_2_ is the dose of drug 2 alone that affects a system by x%, and (D)_1_ + (D)_2_ are the doses of drug 1 and 2 in a combination that also affect a system by x%. CI values below 1 represent synergism, CI value equal to 1 indicates additive effect (i.e., no interaction), and CI values above 1 point to antagonism.

### 4.10. Statistical Analysis

The results of the experiments were expressed as the means ± standard deviation (SD) from at least three independent experiments. The statistical significance was determined by one-way ANOVA analysis (* *p* < 0.05).

## 5. Conclusions

In conclusion, ethanol extract from the bark of *Madhuca longifolia* was demonstrated to possess significant cytotoxic, proapoptotic, and ROS production-stimulating activities in human melanoma cells. Importantly, when a low concentration of the extract was combined with dacarbazine, at the concentration at which the drug was not active, significant anticancer activity of the mixture was observed. Moreover, the pretreatment of melanoma cells with the studied extract prior to dacarbazine application resulted in the potentiation of their anti-melanoma potency as compared to the simultaneous application. It seemed likely that ethanol extract of *M. longifolia* in some way sensitized human melanoma cells to chemotherapeutic drugs. It can therefore be interesting as a promising source of compounds for prospective combination therapy.

## Figures and Tables

**Figure 1 ijms-25-07220-f001:**
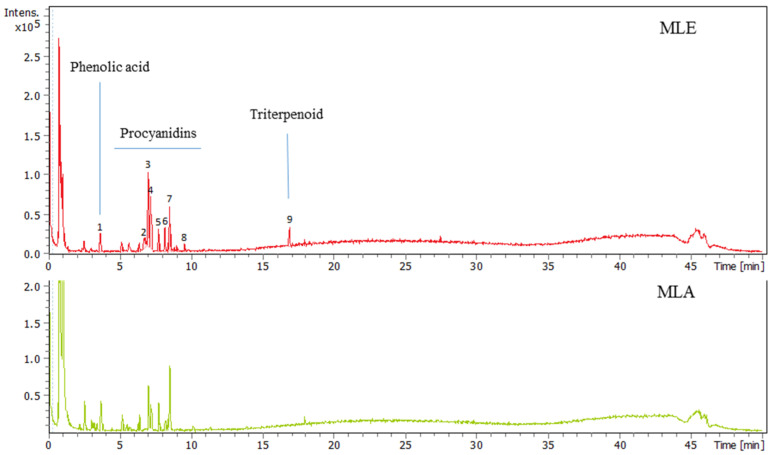
UHPLC-MS chromatograms of the *Madhuca longifolia* ethanol extract (MLE) upper part and aqueous extract (MLA) lower part.

**Figure 2 ijms-25-07220-f002:**
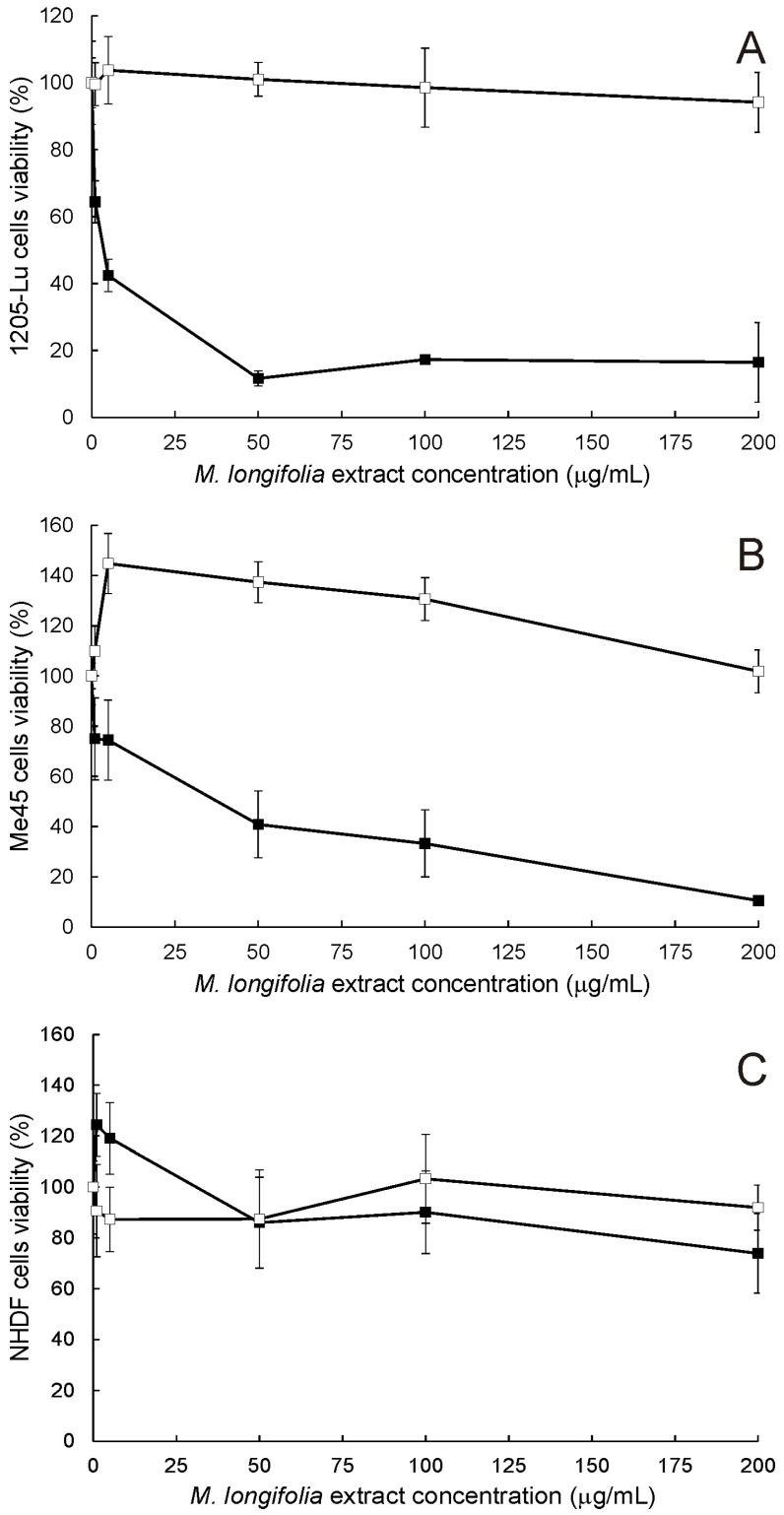
The cytotoxicity assay of MLE (full symbols) and MLA (open symbols) in 1205-Lu (**A**), Me45 (**B**), and NHDF cells (**C**). The means of three experiments ± SD are presented.

**Figure 3 ijms-25-07220-f003:**
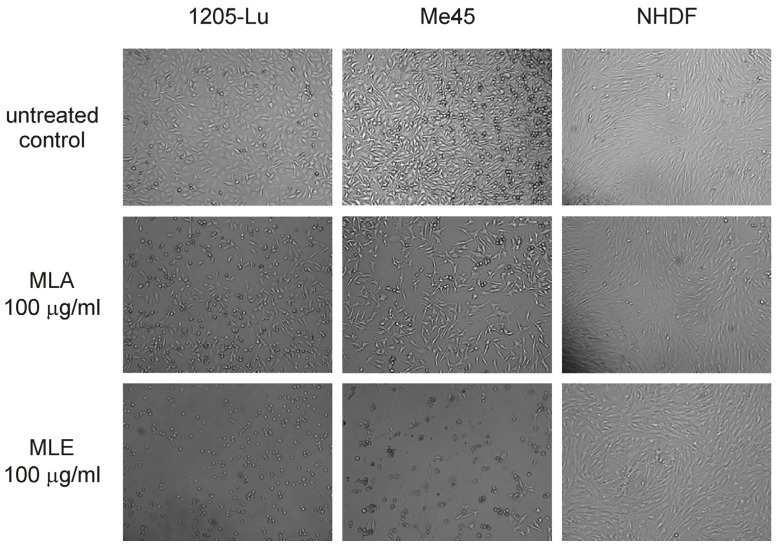
Microscopic images of 1205-Lu (**left**), Me45 (**center**), and NHDF cells (**right**) treated by MLE or MLA at 100 μg/mL for 48 h. Magnification was 10x.

**Figure 4 ijms-25-07220-f004:**
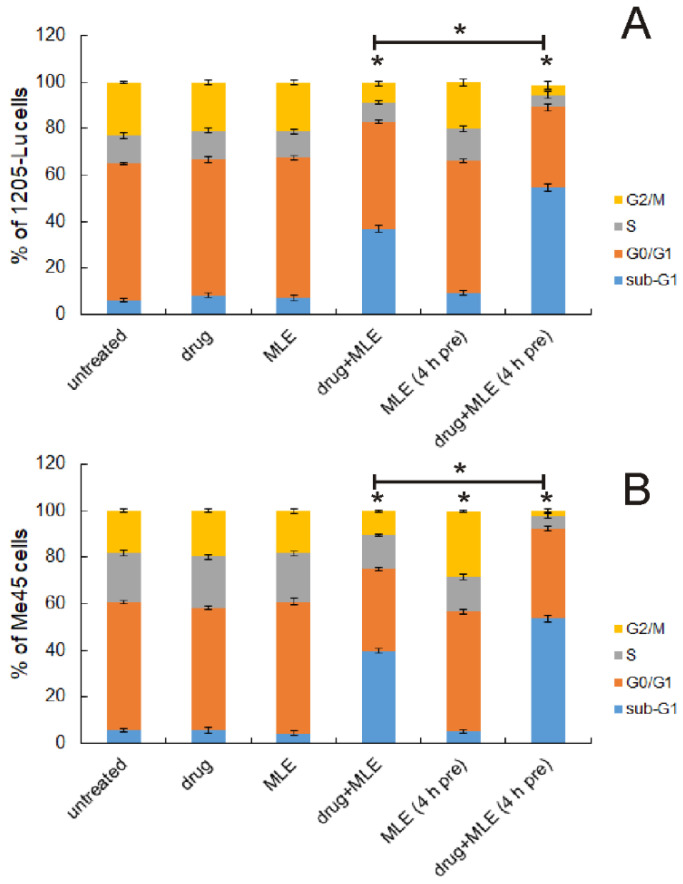
Influence of MLE (at 0.1 μg/mL) on relative distribution of 1205-Lu (**A**) and Me45 cells (**B**) between different cell cycle phases determined on the basis of DNA content. Drug—dacarbazine at 1 ug/mL; 4 h pre—pretreatment of cells by the extract for 4 h before the experiment. Sub-G1 population—dead cells, G0/G1—mononuclear cells, S—DNA replication, G2/M—mitosis. The means of three experiments ± SD are presented (* *p* < 0.05).

**Figure 5 ijms-25-07220-f005:**
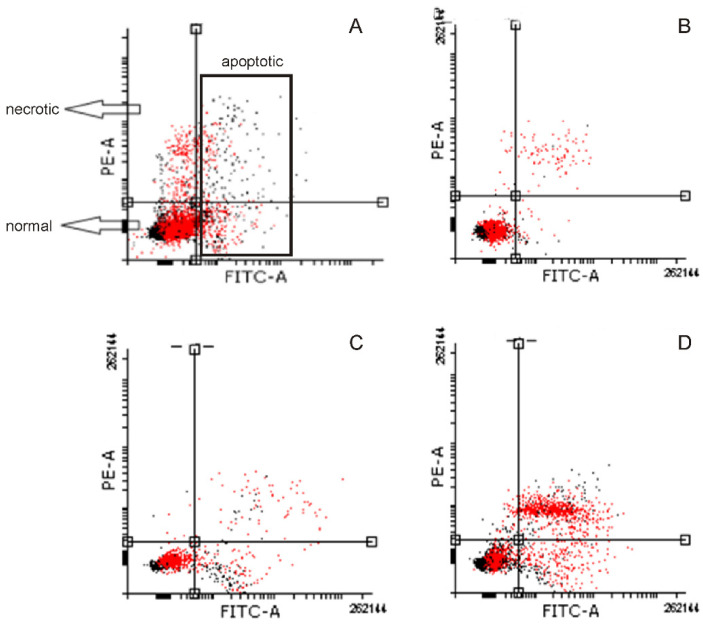
Typical histograms obtained in Me45 cells stained with Annexin-V and PI. Scheme of interpretation of the results (**A**), untreated cells (**B**), cells treated with MLE at IC_10_ (**C**), and cells treated with MLE at IC_50_ (**D**).

**Figure 6 ijms-25-07220-f006:**
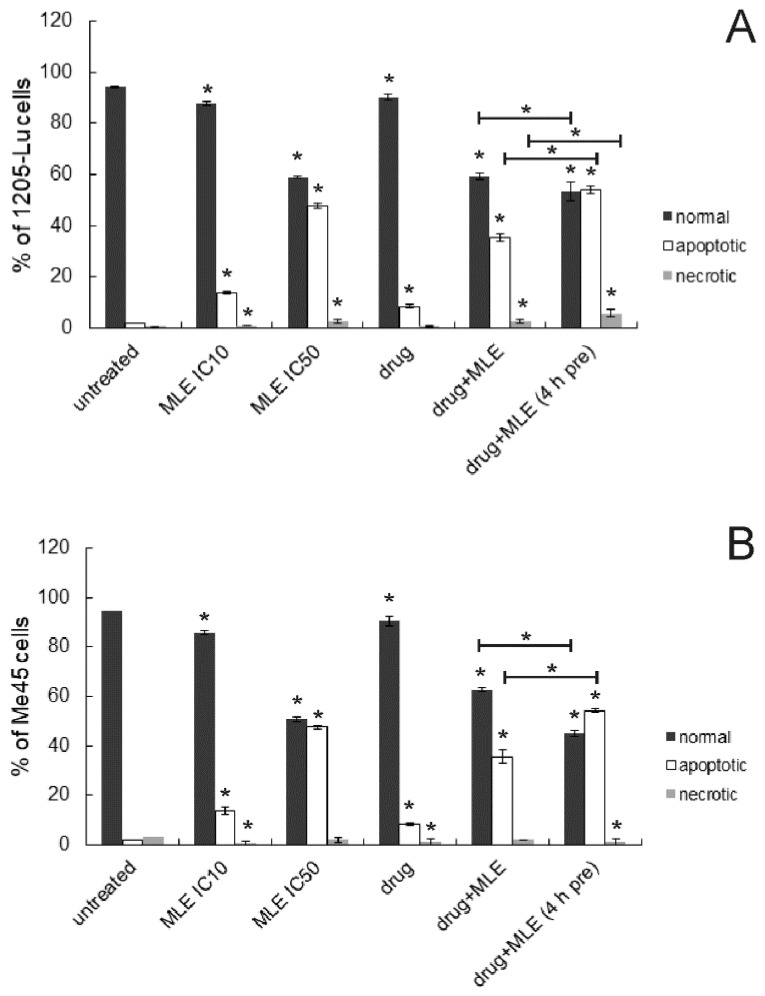
Influence of MLE on the proportion of normal, apoptotic, and necrotic cell populations in 1205-Lu (**A**) and Me45 cells (**B**). MLE at its IC_10_ was used in mixtures with the drug. Cells were recognized as viable (Annexin-V and PI negative), apoptotic (Annexin-V positive and PI negative), and necrotic (Annexin-V and PI positive) based on the measurement of cell-associated fluorescence of FITC-Annexin-V conjugate and PI. Drug—dacarbazine at 1 ug/mL; 4 h pre—pretreatment of cells by the extract for 4 h before the experiment. The means of three experiments ± SD are presented (* *p* < 0.05).

**Figure 7 ijms-25-07220-f007:**
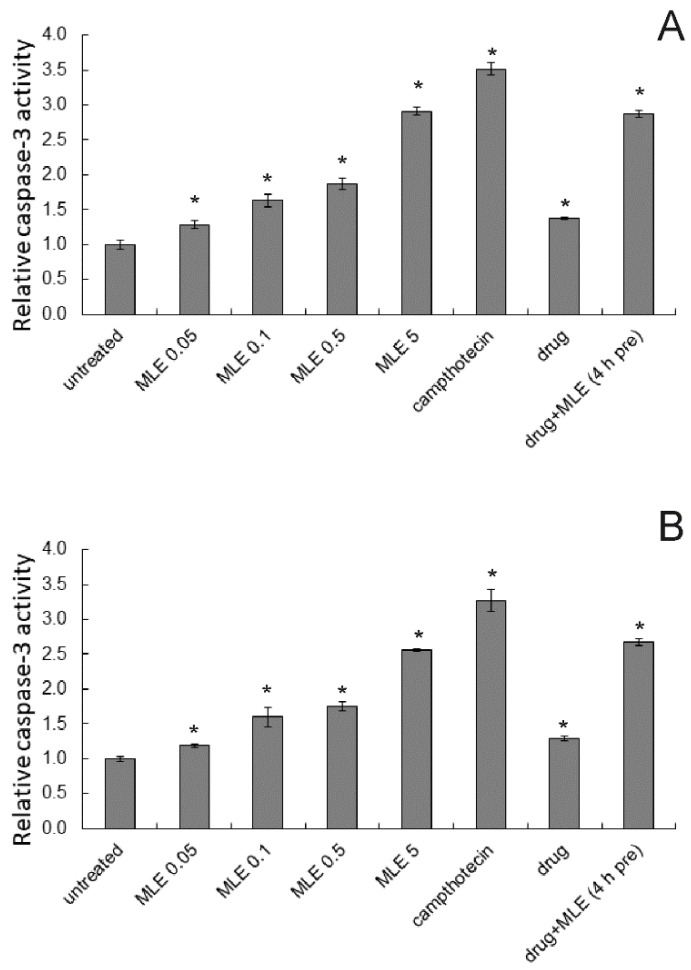
Influence of MLE in different concentrations (in μg/mL) on relative activity of caspase-3 in 1205-Lu (**A**) and Me45 cells (**B**). Campthotecin at 10 μM was used as a positive control. MLE at 0.1 μg/mL was used in mixtures with the drug. Drug—dacarbazine at 1 ug/mL; 4 h pre—pretreatment of cells by the extract for 4 h before the experiment. The means of three experiments ± SD are presented (* *p* < 0.05).

**Figure 8 ijms-25-07220-f008:**
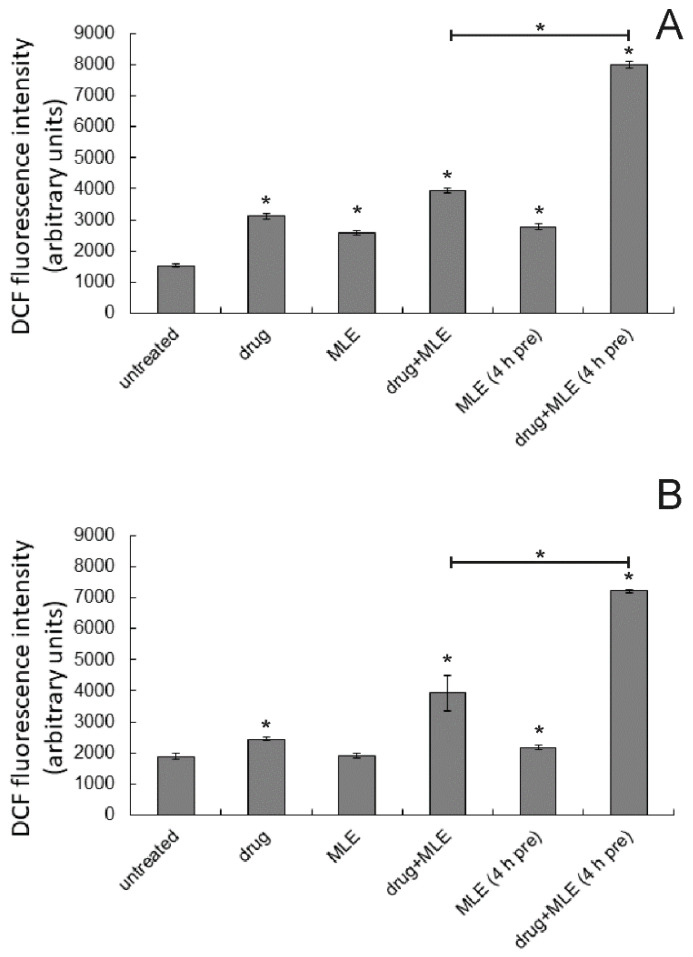
ROS level in 1205-Lu (**A**) and Me45 cells (**B**) treated by MLE (0.1 μg/mL) for 48 h. Drug—dacarbazine at 1 ug/mL; 4 h pre—pretreatment of cells by the extract for 4 h before the experiment. The means of three experiments ± SD are presented (* *p* < 0.05).

**Table 1 ijms-25-07220-t001:** Combination of MLE with dacarbazine against caspase 3 activation.

Cell Line	Concentration (μg/mL)		Ratio	Combination Index (CI)
	MLE	Dacarbazine		
	0.05	1	1:20	0.976
1205-Lu	0.10	1	1:10	0.654
	0.50	1	1:2	0.892
	5.00	1	5:1	1.345
	0.05	1	1:20	0.954
Me45	0.10	1	1:10	0.695
	0.50	1	1:2	0.912
	5.00	1	5:1	1.327

Dose and effect data were obtained from the caspase-3 assay (mean values of three experiments) and analyzed by CompuSyn software. CI values were calculated by CompuSyn software. CI = 1 indicates additive effect, CI < 1—synergism, and CI > 1—antagonism.

## Data Availability

Data are available from the authors K.Ś.-P., A.B., M.G., M.S., A.P.-Ł., and O.W.

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
