# Peer review of "Pretreatment of Melanoma Cells with Aqueous Ethanol Extract from Madhuca longifolia Bark Strongly Potentiates the Activity of a Low Dose of Dacarbazine"

_ijms, 2024, doi:10.3390/ijms25137220_

Round 1

Reviewer 1 Report

Comments and Suggestions for Authors

The present research article provides information regarding the biological activity of an ethanol extract of Madhuca longifolia Bark against melanoma cells. Introduction is well written regarding its content, as well as materials and methods section is well described.

In general, the aim of the study is welcome. However, the manuscript as it is, has some serious drawbacks and I do not recommend its acceptance until are solved.

General Comments

Editing of the English language is required throughout the text. Please make the necessary changes (syntax changes, correct use of tenses, expression errors)

Title

The title of the manuscript discusses an ethanol extract. However, when you describe the extraction method you followed (section 2.3, line 85), you write that extraction was performed using ethanol 70% (v/v). Which other solvent was used for the remaining 30% (v/v)? Please clarify whether you used only ethanol or a system solvent to extract secondary metabolites and please correct accordingly throughout the text.

 Results

Section 3.1 Analysis of Madhuca longifolia extracts

1.       Ms/ms identification for peak 1, peak 7, peak 8 is missing. Generally, you can’t base only on ms analysis for the correct identification of a compound unless you use a standard compound to support your results.

2.       Line 183: Peak 1: “This formula was in good agreement with hydroxybenzoic acid derivatives such as p-hydroxybenzoic acid or salicylic acid”. Which is the compound presented in your extract? You should select one.

3.       Line 186: Peak 2: “Additionally ion at m/z 865.1994 of smaller intensity was observed”. Is this a fragment ion derived from the deprotonated molecule  [M-H]- at m/z 577.1357? Or it is a precursor deprotonated ion? If this so, which product ions afford?

4.        Line 195: Peak 4: “suggesting the presence of catechin/epicatechin molecule”. Catechin and epicatechin are epimers with different biological activity. You should specify which molecule is present in your extract. In addition, for the deprotonated ions at m/z 577.1357 and 865.2004, please provide the respective fragment ions.

5.       Line 200: Peak 5: please provide m/z fragment ions

6.       Line 202: Peak 6: which are those fragment ions that make you suggest the presence of epiafzelechin/afzelechin and epictechin/catechin units? Similar fragment ions you found also regarding compounds 2 and 3, however no further comment regarding the presence of specific units is made

7.       Line 207: peak 7: “was very similar”. What do you mean? What made you conclude this? Provide the corresponding reference. In addition, the pseudomolecular ion [M − H]− at m/z 425.0763, had the same molecular formula as epiafzelechin gallate? So, you want to say that experimental and theoretical masses of these 2 compounds is above a certain limit therefore compound 7 is not epiafzelechin gallate? Please rephrase lines 207-210 by using the appropriate terminology.

Generally, identification of the compounds is very ambiguous. No standard solutions were used to confirm the presence of some compounds. Ms/ms identification is not correctly presented since no explanation regarding the presence of specific fragment ions is presented by the authors. Neither a relative reference to support the results is provided.

Discussion section

Line 352: “the presence of triterpenoids”. The list of identified compounds includes only one triterpenoid. Please correct.

Lines 350-353: The study of Bürkel et al., regarding the biological activity, deals with the antibacterial and antidiabetic activity of Madhuca longifolia. I think that your statement written at lines 350-353 is quite risky. You should search for literature data that correlate the biological activity you studied with the compounds presented in your extract.

Lines 353-356: What is written here is right, since many times extracts exert stronger biologic activity than an isolated compound. However, I find a contradiction with what is previously written at lines 352-353. Please correct. Otherwise, I think that by removing the phrase “on the other hand”-line 353, the contradiction maybe disappears. Please clarify the meaning.

Line 373: “0.1 μg/mL”. Please confirm if the units are correctly presented.

Conclusions

Line 417-418: Please remove the word “non-toxic”. Obviously, the concentration used should be non-toxic. Otherwise, please explain better the use of this word

Comments on the Quality of English Language

Moderate editing of english language is required

Author Response

Revewer 1

The present research article provides information regarding the biological activity of an ethanol extract of Madhuca longifolia Bark against melanoma cells. Introduction is well written regarding its content, as well as materials and methods section is well described.

In general, the aim of the study is welcome. However, the manuscript as it is, has some serious drawbacks and I do not recommend its acceptance until are solved.

General Comments

Editing of the English language is required throughout the text. Please make the necessary changes (syntax changes, correct use of tenses, expression errors)

Title

The title of the manuscript discusses an ethanol extract. However, when you describe the extraction method you followed (section 2.3, line 85), you write that extraction was performed using ethanol 70% (v/v). Which other solvent was used for the remaining 30% (v/v)? Please clarify whether you used only ethanol or a system solvent to extract secondary metabolites and please correct accordingly throughout the text.

In the present work, we have studied two Madhuca longifolia bark extracts: 1) MLA which was extracted with hot distilled water, and 2) MLE which was extracted with ethanol:water mixture (70%/30%). The title of the article was changed and it was described in detail in Materials and methods section. However, we decided to leave the term “ethanol extract” throughout the text for the sake of clarity and to differentiate MLE and MLA precisely.

Results

Section 3.1 Analysis of Madhuca longifolia extracts

  1. Ms/ms identification for peak 1, peak 7, peak 8 is missing. Generally, you can’t base only on ms analysis for the correct identification of a compound unless you use a standard compound to support your results.

The detailed description of all peaks has been included into 3.1. Section of the Results. Supplementary material with more detailed spectra has been also added for clarity .
We agree with the Referee that it is impossible to give an accurate identification of a compound by MS spectrometry only, especially if there is a need to distinguish between isomers or epimers, like in the case of procyanidins, prior their isolation and NMR structure elucidation. Since procyanidins are a group of natural compounds characterized by high degree of polymerization, based on the literature and the MS data we were able to name the investigated compounds
tentatively. Therefore at the beginning of this section (3.1) we stressed that “These peaks were tentatively characterized based on their MS and MS/MS spectra.”        
In our study, the identification of compounds was achieved by comparing the obtained data with the established literature patterns. This approach ensured the accuracy and reliability of our results. Here is the detailed explanation of our methodology: 1. **Data Acquisition:** - We utilized advanced analytical techniques such as mass spectrometry (HRMS), 2. **Literature Comparison:** - The spectral data obtained from our experiments were meticulously compared with reference data from peer-reviewed scientific literature. This included published spectral libraries and databases known for their comprehensiveness and reliability. Specific attention was given to key spectral features such as peak positions, relative intensities, and fragmentation patterns, which are characteristic of the compounds being studied. 3. **Validation:** - Any discrepancies were carefully examined, and compounds were only confirmed when a clear match with literature data was established. 4. **Conclusion:** - We believe that this systematic comparison with literature data enhances the credibility of our results and aligns with best practices in compound identification. We hope this explanation clarifies our methodology and addresses any concerns regarding the identification process

  1. Line 183: Peak 1: “This formula was in good agreement with hydroxybenzoic acid derivatives such as p-hydroxybenzoic acid or salicylic acid”. Which is the compound presented in your extract? You should select one.

MS spectrometry can support the identification of the compounds However, when structural isomers have identical fragmentation pathways, then, even with high-mass resolution, mass spectrometry alone cannot distinguish these isomers under conventional ion activation conditions. Since p-hydroxybenzoic acid or salicylic acid are isomers it was impossible to differentiate the two basing on the obtained data.

  1. Line 186: Peak 2: “Additionally ion at m/z 865.1994 of smaller intensity was observed”. Is this a fragment ion derived from the deprotonated molecule [M-H]- at m/z 577.1357? Or it is a precursor deprotonated ion? If this so, which product ions afford?

Procyanidins due to their nature are difficult to separate chromatographically and sometimes in one peak there are two or three types of procyanidins present. In this case we had two pseudomolecular ions [M-H]- at m/z 577.1357 ( assigned to procyanidin dimer) and  m/z 865.1994 (assigned to procyanidin trimer).

  1. Line 195: Peak 4: “suggesting the presence of catechin/epicatechin molecule”. Catechin and epicatechin are epimers with different biological activity. You should specify which molecule is present in your extract. In addition, for the deprotonated ions at m/z 577.1357 and 865.2004, please provide the respective fragment ions.

We have included the supplementary data with MS and MS/MS spectra of the investigated peaks.

  1. Line 200: Peak 5: please provide m/z fragment ions

We have included the supplementary data with MS and MS/MS spectra of the investigated peaks.

  1. Line 202: Peak 6: which are those fragment ions that make you suggest the presence of epiafzelechin/afzelechin and epictechin/catechin units? Similar fragment ions you found also regarding compounds 2 and 3, however no further comment regarding the presence of specific units is made

When writing this fragment we based on supplementary materials included in the publication of Bürkel et al. (Heliyon 2023, 9, e21134), where this compound was fully characterized. However following the advice of the Referee we decided to keep the same format for all procyanidins so that we removed the mentioned fragment.

  1. Line 207: peak 7: “was very similar”. What do you mean? What made you conclude this? Provide the corresponding reference. In addition, the pseudomolecular ion [M − H]− at m/z 425.0763, had the same molecular formula as epiafzelechin gallate? So, you want to say that experimental and theoretical masses of these 2 compounds is above a certain limit therefore compound 7 is not epiafzelechin gallate? Please rephrase lines 207-210 by using the appropriate terminology.

The description of  peak 7 has been changed. We have revised the mass spectra and assigned this peak to be 3,4,5-trimetoxyphenyl 1-O-β-(6-sulpho)-glucopyranoside. Our finding is supported by the reference (Maldini et al. Phytochemistry 70 (2009) 641–649).
Considering the presence of procyanidins in the investigated plant material, we conclude that our MS results are consistent with the general data of procyanidins (Lin et al. J. Agric. Food Chem. 2014, 62, 9387−9400, and
Bürkel et al. (Heliyon 2023, 9, e21134).

Generally, identification of the compounds is very ambiguous. No standard solutions were used to confirm the presence of some compounds. Ms/ms identification is not correctly presented since no explanation regarding the presence of specific fragment ions is presented by the authors. Neither a relative reference to support the results is provided.

Thank you for your insightful comments. However, we would like to take the opportunity to elucidate the objectives and the scope of our project more clearly . The identification of the extract components is indeed a crucial aspect of our research, and it forms the foundation of our ongoing studies. The accurate identification of compounds not only ensures the reproducibility and reliability of our findings but also enhances our understanding of the active components and their interactions.
However, we need to stress that the present project specifically aimed to identify the plant extract able to augment the activity of dacarbazine. Dacarbazine is a well-known chemotherapeutic agent, and enhancing its efficacy could have significant therapeutic implications. In this project, our primary focus was on identifying natural extracts that could potentiate the anticancer activity of the known drug.
We hope this explanation clarifies the specific objectives and scope of the current project and how it fits within the larger context of our ongoing research.

Discussion section

Line 352: “the presence of triterpenoids”. The list of identified compounds includes only one triterpenoid. Please correct.

The sentence has been modified according to the Referee’s suggestion.

Lines 350-353: The study of Bürkel et al., regarding the biological activity, deals with the antibacterial and antidiabetic activity of Madhuca longifolia. I think that your statement written at lines 350-353 is quite risky. You should search for literature data that correlate the biological activity you studied with the compounds presented in your extract.

The paper of Burkel at al. is the only one (according to our best knowledge) that reports the isolation and identification of procyanidins from Madhuca longifolia bark (see the supplementary data of this publication). According to the Referee’s suggestion the statement from the lines 350-353 was strengthened by additional references.

Lines 353-356: What is written here is right, since many times extracts exert stronger biologic activity than an isolated compound. However, I find a contradiction with what is previously written at lines 352-353. Please correct. Otherwise, I think that by removing the phrase “on the other hand”-line 353, the contradiction maybe disappears. Please clarify the meaning.

The last sentence of the paragraph has been rephrased to avoid ambiguity.

Line 373: “0.1 μg/mL”. Please confirm if the units are correctly presented.

The units have been checked and corrected. There was an error in the caption of Fig. 8.

Conclusions

Line 417-418: Please remove the word “non-toxic”. Obviously, the concentration used should be non-toxic. Otherwise, please explain better the use of this word

The statement has been rephrased according to the Referee’s suggestion.

Comments on the Quality of English Language

The language of the manuscript has been thoroughly read, corrected and revised where necessary.

Reviewer 2 Report

Comments and Suggestions for Authors

Wesołowska and colleagues deliver an original research article on the in vitro anticancer effects of polar extracts obtained from Madhuca longifolia bark, as well as the potential benefits of combining them with dacarbazine. For this purpose, authors selected 1205-Lu and Me45 cells as models of melanoma and evaluated the potential pro-apoptotic effects and interference on ROS production. Identification of bioactives was attempted by means of UHPLC-ESI-MS.

While the concept of the manuscript fits within the scope of IJMS, scientific outcomes have limited relevance and the novelty lies solely in the cytotoxic effects on melanoma cell lines. Major issues are stressed out below:

A. The qualitative chemical characterization of the extracts is preliminary and does not enable a conclusive identification (Section 3.1.). For example, authors refer to the identification of procyanidin dimers and trimers but the MS/MS data would enable a detailed characterization of procyanidin types. It should also be noted that MS/MS data should be made available in the manuscript. Furthermore, since the aqueous and ethanol extracts appear to share the same qualitative profile but differ on their in vitro effects, authors should carry out a quantitative chemical characterization to further elucidate the contribution of each identified constituent and/or to conclude that the effects are mainly attributed to compounds from other structural classes that were not identified.

B. Authors are requested to provide further details on the processing of the plant material and extraction procedures to ensure standardization and future repeatability of their findings. Specifically, the collection date of the plant material should be indicated, along with the mean particle size of the powdered sample, authors being also requested to indicate the duration of extraction.

C. In section 3.2., authors indicate that the extracts were used in a concentration range between 25 and 200 mg/mL which appears to be a typo as such concentrations have no biological relevance. Furthermore, as seen in Figure 2.C., it seems unlikely that the extracts did not impact the cell viability of NHDF cells, especially upon exposure to the ethanol extract. It should also be noted that the use of the Student t-test for the statistical analysis of cell-based assays is not appropriate as there are several test groups (different concentrations) and the cell population is to be considered an additional variable. In this specific matter, the use of ANOVA would be more appropriate.  

D. Authors should also attempt to reanalyse the in vitro data on the combination of Madhuca longifolia bark extracts with dacarbazine, and to determine their interaction (e.g., addictive effect? Synergistic effect?) through the combination index or sobologram analyses.

E. Some inaccurate terms should be revised as in:

Lines 43 and 346-347: While some specific compounds are (or are reported to be) active i.e., displaying pharmacological or biological effects, authors broadly refer to structural classes of secondary metabolites that cannot be globally ascribed as active.

Line 47: What do the authors mean with a “dynamic growth” of melanoma incidence?

Comments on the Quality of English Language

While generally well-written, the manuscript still requires revision of the writing style and correction of some typos. A few examples demonstrating this can be found in lines 52-55, 148-149, line 198 (revise “porcyanidins” to “procyanidins”), and 371-372. Furthermore, Sapotaceae (lines 36-37) is not to be italicized as it refers to the family, while when using the abbreviation of “para” should be italicized, e.g., “p-nitroanilide” (line 170) or “p-hydroxybenzoic acid” (line 185).

Author Response

Reviewer 2

Wesołowska and colleagues deliver an original research article on the in vitro anticancer effects of polar extracts obtained from Madhuca longifolia bark, as well as the potential benefits of combining them with dacarbazine. For this purpose, authors selected 1205-Lu and Me45 cells as models of melanoma and evaluated the potential pro-apoptotic effects and interference on ROS production. Identification of bioactives was attempted by means of UHPLC-ESI-MS.

While the concept of the manuscript fits within the scope of IJMS, scientific outcomes have limited relevance and the novelty lies solely in the cytotoxic effects on melanoma cell lines. Major issues are stressed out below:

  1. The qualitative chemical characterization of the extracts is preliminary and does not enable a conclusive identification (Section 3.1.). For example, authors refer to the identification of procyanidin dimers and trimers but the MS/MS data would enable a detailed characterization of procyanidin types. It should also be noted that MS/MS data should be made available in the manuscript. Furthermore, since the aqueous and ethanol extracts appear to share the same qualitative profile but differ on their in vitro effects, authors should carry out a quantitative chemical characterization to further elucidate the contribution of each identified constituent and/or to conclude that the effects are mainly attributed to compounds from other structural classes that were not identified.

The detailed description of all peaks has been included into 3.1. Section of the Results. Supplementary material with more detailed spectra has been also added for clarity  .
We agree with the Referee that it is impossible to give an accurate identification of a compound by MS spectrometry only, especially if there is a need to distinguish between isomers or epimers, like in the case of procyanidins, prior their isolation and NMR structure elucidation. Since procyanidins are a group of natural compounds characterized by high degree of polymerization, based on the literature and the MS data we were able to name the investigated compounds tentatively. Therefore at the beginning of this section (3.1) we stressed that “These peaks were tentatively characterized based on their MS and MS/MS spectra.”         
In our study, the identification of compounds was achieved by comparing the obtained data with the established literature patterns. This approach ensured the accuracy and reliability of our results. Here is the detailed explanation of our methodology: 1. **Data Acquisition:** - We utilized advanced analytical techniques such as mass spectrometry (HRMS), 2. **Literature Comparison:** - The spectral data obtained from our experiments were meticulously compared with reference data from peer-reviewed scientific literature. This included published spectral libraries and databases known for their comprehensiveness and reliability. Specific attention was given to key spectral features such as peak positions, relative intensities, and fragmentation patterns, which are characteristic of the compounds being studied. 3. **Validation:** - Any discrepancies were carefully examined, and compounds were only confirmed when a clear match with literature data was established. 4. **Conclusion:** - We believe that this systematic comparison with literature data enhances the credibility of our results and aligns with best practices in compound identification. We hope this explanation clarifies our methodology and addresses any concerns regarding the identification process.           
As to the chemical composition differences between MLE and MLS that mirror in quite differential in vitro activity of the two extracts we can state that d
efinitely there was no triterpenoid peak in the aqueous extract and content of procyanidins was higher in the ethanol extract based on the MS peak area. However we agree that the next step should be fractionation, isolation and full structure elucidation of the active ingredients.   
However, the full identification of extracts’ components  was slightly beyond of the scope of the present paper which specifically aimed to identify the plant extract able to augment the activity of dacarbazine. Dacarbazine is a well-known chemotherapeutic agent, and enhancing its efficacy could have significant therapeutic implications. In this project, our primary focus was on identifying natural extracts that could potentiate the anticancer activity of the known drug.  

  1. Authors are requested to provide further details on the processing of the plant material and extraction procedures to ensure standardization and future repeatability of their findings. Specifically, the collection date of the plant material should be indicated, along with the mean particle size of the powdered sample, authors being also requested to indicate the duration of extraction.

The appropriate section of Materials and Methods has been modified to give the requested details.

  1. In section 3.2., authors indicate that the extracts were used in a concentration range between 25 and 200 mg/mL which appears to be a typo as such concentrations have no biological relevance. Furthermore, as seen in Figure 2.C., it seems unlikely that the extracts did not impact the cell viability of NHDF cells, especially upon exposure to the ethanol extract. It should also be noted that the use of the Student t-test for the statistical analysis of cell-based assays is not appropriate as there are several test groups (different concentrations) and the cell population is to be considered an additional variable. In this specific matter, the use of ANOVA would be more appropriate.

Thank you for this valuable comment. Of course this was the mistake. Undoubtedly, we worked in the range of 25 and 200 g/mL. It seemed that we had a global problem with formatting the Greek letter “mi” throughout the text it was carefully revised.
Regarding the second part of the reviewer's comment, we actually included both healthy and cancer cell lines in our work. As rightly noted by the reviewer in Figure 2.C., the extracts show a slight impact on the cell viability of NHDF cells. in accordance with the Reviewer's comments, appropriate changes have been made to the text. Nevertheless, by comparing the effect of the tested extracts on both types of cells, we wanted to check whether they have a preferential effect on cancer cells. By showing the differences in growth between healthy and cancer cells that were exposed to extracts in the same concentration range, we wanted to pay the attention on the fact that extracts inhibited the growth of healthy cells much less than cancer cells. This selectivity is crucial in developing targeted therapies that specifically inhibit the growth of cancer cells while minimizing the harm to healthy cells.
According to the Reviewer’s suggestion we re-analyzed statistical significance of the obtained results using ANOVA analysis. The results obtained by this method did not show any differences to our previous analysis.

  1. Authors should also attempt to reanalyse the in vitro data on the combination of Madhuca longifolia bark extracts with dacarbazine, and to determine their interaction (e.g., addictive effect? Synergistic effect?) through the combination index or isobologram analyses.

The combination index analysis of the obtained results has been performed and the synergy was confirmed between MLE and dacarbazine. The new Table 1 and the appropriate comments have been introduced to the text (Materials and Methods, Results, Discussion sections).

  1. Some inaccurate terms should be revised as in:

Lines 43 and 346-347: While some specific compounds are (or are reported to be) active i.e., displaying pharmacological or biological effects, authors broadly refer to structural classes of secondary metabolites that cannot be globally ascribed as active.

The sentence in lines 346-47 has been changed not to suggest that all e.g. triterpenoids are biologically active.

Line 47: What do the authors mean with a “dynamic growth” of melanoma incidence?

The statement has been rephrased to avoid a non-precise statement.

Comments on the Quality of English Language

While generally well-written, the manuscript still requires revision of the writing style and correction of some typos. A few examples demonstrating this can be found in lines 52-55, 148-149, line 198 (revise “porcyanidins” to “procyanidins”), and 371-372. Furthermore, Sapotaceae (lines 36-37) is not to be italicized as it refers to the family, while when using the abbreviation of “para” should be italicized, e.g., “p-nitroanilide” (line 170) or “p-hydroxybenzoic acid” (line 185).

The English of the paper has been thoroughly revised.

Round 2

Reviewer 1 Report

Comments and Suggestions for Authors

Authors have adequately addressed my comments/suggestions.

I have no further comments